# Association between age at first birth and risk of rheumatoid arthritis in women: Evidence from NHANES 2011–2020

Hang Cui[1,2°], Chenyang Huang[1,2°], Yuntian Ye[1,2], Tianci Guo[1,2], Weijie Yu[1,2], Puyu Niu[1,2], Kuiliang Gao[1,2], Jiajin Liu[1,2], Aifeng Liu[1,2]*

1 First Teaching Hospital of Tianjin University of Traditional Chinese Medicine, Tianjin, China, 2 National Clinical Research Center for Chinese Medicine Acupuncture and Moxibustion, Tianjin, China

° These authors contributed equally to this work.

* draifeng@163.com

**Data Availability Statement:** Data availability statement The continuous National Health and Nutrition Examination Survey (NHANES) dataset are publicly available at the National Center for

## Abstract

### Objective

This study focused on investigating the relation of age at first birth (AFB) with rheumatoid arthritis (RA) risk in women based on the 2011–2020 NHANES (National Health and Nutrition Examination Survey) data.

### Methods

Women were analyzed using National Health and Nutrition Examination Survey (NHANES) data from 2011 to 2020 in the US. Both AFB and RA diagnoses were obtained through self-reported questionnaires. Odds ratios (ORs) and 95% confidence intervals (CIs) were determined using logistic regression models.

### Results

Among the 7,449 women included in this study, 552 (7%) were diagnosed with RA. In comparison with women who had an AFB of 30–32 years (reference group), those who had an AFB < 18, 18–20, 21–23, 24–26, and > 35 years had the fully adjusted ORs and 95% CIs of 4.00 (95% CI 1.70, 9.40), 2.90 (95% CI 1.25, 6.73), 3.00 (95% CI 1.32, 6.80), 3.18 (95% CI 1.36–7.42), and 3.36 (95% CI 1.04–10.7), respectively. Due to the limitations inherent in cross-sectional studies, we have not observed significant differences in the risk of RA between women aged 27–29 and 33–35 at the AFB. Further research is warranted to refine these findings.

### Conclusion

Women with an AFB < 26, or > 35 years have a higher risk of developing RA later in life. Policymakers may consider focusing more on women in these AFB age ranges in screening RA and making preventive measures.

Health Statistics of the Center for Disease Control and Prevention and the link to the database is https://www.cdc.gov/nchs/nhanes/index.htm.

**Funding:** Funding 1.National Natural Science Foundation of China General Program (82474538);2.Tianjin Municipal Health Commission Jinmen Medical Talents Program (TJSJMYXYC-D2-028);3.Tianjin Science and Technology Plan Project (23KPXMRC00170).

**Competing interests:** The authors have declared that no competing interests exist.

## Introduction

Rheumatoid arthritis (RA) represents the frequently seen autoimmune disease with the typical features of symmetrical polyarthritis and chronic synovial inflammation, leading to progressive joint destruction. The key pathological changes in RA include chronic synovitis and pannus formation, which can erode cartilage and bone, resulting in joint damage. RA shows a global prevalence of 0.5%-1%, and the female-to-male ratio is approximately 4:1 [1,2]. Currently, the exact etiology of RA remains unclear, but it may be associated with infections, genetics, estrogen levels, and environmental factors like cold and humidity. Moreover, physical exertion, malnutrition, trauma, and psychological stress are also the potential triggers. Early diagnosis and prevention are key factors for managing RA, and identifying the RA risk factors can help inform public health policies to address this challenge.

The RA prevalence varies by gender, and women are 4–5 times more likely to develop RA than men aged <50 years, while about twice as likely as men aged 60–70 years [3,4]. Women's health is influenced by various reproductive factors, which may be important for RA occurrence. Factors including age at menarche, age at first birth (AFB), hysterectomy or oophorectomy history, and menopause are closely associated with RA prevalence [5].

Among the numerous influencing factors, pregnancy accounts for a key determinant of female health [6]. Research has indicated that during pregnancy, maternal hormone levels and immune function undergo significant changes [7]. However, no previous studies have conducted a detailed exploration of the relationship between AFB and the risk of RA. Based on previous research regarding the factors influencing RA in women, we hypothesize that AFB may be associated with RA. Therefore, this study aimed to use NHANES (2011–2020) data for investigating the relationship of AFB with RA risk, so as to provide more comprehensive evidence from a reproductive perspective.

## Materials and methods

### Study subjects

We obtained data based on NHANES (National Health and Nutrition Examination Survey), which surveys approximately 5,000 individuals annually from across the United States. The NHANES database covers demographic, dietary, laboratory, examination, questionnaire, as well as restricted access data. Detailed information on the design and data of NHANES can be obtained from https://www.cdc.gov/nchs/nhanes/.

The NHANES is a complex, stratified, multistage, probability-cluster program designed to assess Americans' health and nutritional status. The 2011–2020 NHANES data were included in this work for analysis, comprising 62,662 participants over the five survey cycles. The following groups were excluded: (1) males (n = 31,005), (2) individuals with missing AFB data (n = 9,356), (3) individuals with missing RA data (n = 8,569), and (4) individuals with erroneous covariate data (n = 6,283). Finally, 7,449 women were enrolled for analysis (Fig 1).

### Outcome definition

The primary outcome in this study was RA, with the diagnosis being obtained through a self-reported questionnaire by the question (MCQ160A): "Has a doctor or other health professional ever told you that you have arthritis?", and the response of "Yes" or "No" could be included. If the response was "Yes," RA was further assessed through the question (MCQ195): "What type of arthritis is it?", and the responses could be "Rheumatoid arthritis," "Osteoarthritis," "Psoriatic arthritis," "Other," "Refused," and "Don't know." Early studies have suggested a

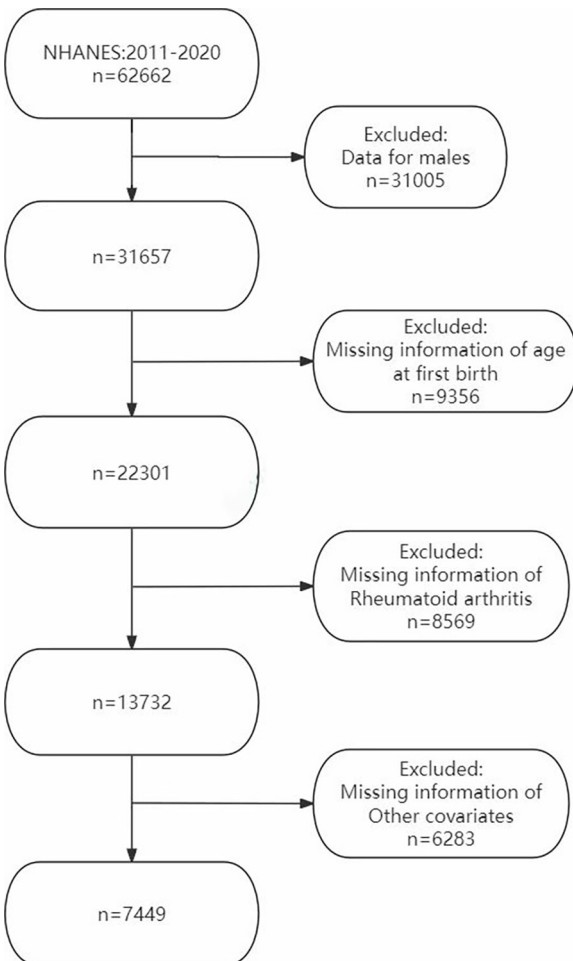

**Fig 1. Flow chart of participants.** From: Association between age at first birth and risk of rheumatoid arthritis in women: Evidence from NHANES 2011–2020.

high level of agreement between clinically diagnosed arthritis and self-reported counterpart [8].

## Reproductive factors

Self-reported questionnaires were used to collect reproductive factors. Subjects answered questions on AFB through the reproductive health questionnaire. Additional data on contraceptive use, hormone therapy, gynecological surgical history (such as oophorectomy or hysterectomy), and information on menopausal status were also collected. Noteworthily, AFB referred to the age when giving the first live birth, excluding miscarriages or stillbirths.

## Covariates

Demographic data were collected through self-reported questionnaires at the time of enrollment. Covariates probably affecting RA were age, race/ethnicity (Mexican American, other Hispanic, non-Hispanic White, non-Hispanic Black, other race including multiracial) [9], education level (< high school, high school graduate, > high school), poverty-income ratio (PIR ≤ 1.31, 1.31–3.38, ≥ 3.39), body mass index (BMI <25, 25–30, ≥30) [10], alcohol

consumption (yes/no) [11], smoking status (yes/no) [12], hypertension (yes/no) [13], diabetes (yes/no), menopausal status (yes/no) [14], use of hormone therapy (yes/no) [15], use of oral contraceptive (yes/no) [16], hysterectomy history (yes/no) [17], bilateral oophorectomy history (yes/no) [18], uric acid (UA), and total cholesterol (TC).

## Statistical analysis

STATA version 18.0 was employed for statistical analyses due to the complicated NHANES design and Mobile Examination Center (MEC) examination sampling weights. A P-value < 0.05 stood for statistical significance. The closest value of categorical data was used to impute missing data. Descriptive statistics was our primary analysis, with continuous data being represented by survey-weighted means and standard errors (SE), whereas categorical data being represented by counts (N) and percentages (%). Basic features among groups were compared by Kruskal-Wallis H tests and Pearson's chi-squared tests.

For the primary analysis, we employed multivariable logistic regression models for assessing the relation of AFB with RA. AFB of 30–32 years, which had the least survey-weighted prevalence, was used for reference. We determined odds ratios (ORs) together with associated 95% confidence intervals (CIs) for determining the association strength. No factor was adjusted in Model 1; in Model 2, age and race were adjusted; while in Model 3, age, race, education level, PIR, BMI, hypertension, diabetes, smoking, alcohol consumption, UA, TC, menopausal status (yes/no), use of hormone therapy (yes/no), use of oral contraceptive (yes/no), hysterectomy (yes/no) and bilateral oophorectomy (yes/no) history were adjusted. In the fully adjusted model, restricted cubic spline analysis was conducted for explaining the nonlinear relation of AFB with RA. Spline curves were generated using R version 4.3.2.

We further explored the potential sources of heterogeneity through stratifying the analyses according to hypertension (yes/no), diabetes (yes/no), use of hormone therapy (yes/no), menopausal status (yes/no), smoking (yes/no), and alcohol consumption (yes/no). To test whether the results were robust, we conducted sensitivity analysis through eliminating subjects undergoing gynecological surgeries (hysterectomy or oophorectomy), those with hypertension or diabetes, and those who were postmenopausal.

## Results

Among the 7,449 women enrolled in the present work, 552 (7%) had the diagnosis of RA. Table 1 summarizes baseline subject features according to AFB categories. Subjects were classified as groups according to AFB, including < 18, 18–20, 21–23, 24–26, 27–29, 30–32, 33–35, and > 35 years. Age, blood pressure, education, PIR, smoking status, and menopausal status were significantly different across AFB categories (P < 0.05). Additionally, subjects who had an AFB < 18 years had highest RA prevalence across all age groups, while those with an AFB of 30–32 years had the smallest survey-weighted RA prevalence. Women of the reference group also had the lowest proportion of BMI between ≥ 25 and < 30, the lowest poverty rate based on PIR, and a lower likelihood of hormone therapy use, hysterectomy, and bilateral oophorectomy. Moreover, these subjects also exhibited the lowest prevalence of hypertension and diabetes.

Subjects with an AFB of 30–32 years were assigned to reference group, and those who had an AFB ≤ 29 years showed a higher OR for RA (range 8.04–2.59, P < 0.05) (Table 2). After confounders were adjusted, those observed relations were somewhat attenuated. When sociodemographic data, reproductive factors and lifestyle factors were adjusted, the association in the AFB ≤ 26 years group was weakened but remained significant (Model 3; fully adjusted OR 4.01–3.19, P < 0.05). Nonetheless, no significant difference could be observed for those with

**Table 1. Basic characteristics of participants according to age at first birth.** From: Association between age at first birth and risk of rheumatoid arthritis in women: Evidence from NHANES 2011–2020.

| | | Total (N = 7449) | Age at first birth (years) | | | | | | | | P |
| --- | --- | --- | --- | --- | --- | --- | --- | --- | --- | --- | --- |
| | | | <18 (N = 1122) | 18-20 (N = 2012) | 21-23 (N = 1651) | 24-26 (N = 1081) | 27-29 (N = 781) | 30-32 (N = 424) | 33-35 (N = 237) | >35 (N = 141) | |
| Alcohol intake (%) | NO | 991(13.3) | 139(12.4) | 288(14.3) | 233(14.1) | 164(15.2) | 102(13.1) | 38(9) | 21(8.9) | 14(10) | 0.163 |
| | YES | 3568(47.9) | 548(48.8) | 994(49.4) | 764(46.3) | 481(44.5) | 350(44.8) | 231(54.4) | 115(48.7) | 82(58.4) | |
| | Unknown | 2890(38.8) | 434(38.7) | 730(36.3) | 654(39.6) | 435(40.2) | 329(42.1) | 155(36.6) | 100(42.4) | 44(31.5) | |
| BMI(%) | normal | 2101(28.2) | 200(17.8) | 425(21.1) | 452(27.4) | 316(29.2) | 307(39.3) | 185(43.7) | 85(35.8) | 46(32.4) | <0.001 |
| | overweight | 2145(28.8) | 340(30.3) | 571(28.4) | 489(29.6) | 352(32.6) | 213(27.3) | 100(23.6) | 58(24.6) | 40(28.1) | |
| | obesity | 3196(42.9) | 582(51.9) | 1016(50.5) | 710(43) | 412(38.1) | 261(33.4) | 139(32.7) | 94(39.6) | 56(39.5) | |
| Hypertension (%) | NO | 4901(65.8) | 708(63.1) | 1219(60.6) | 984(59.6) | 733(67.8) | 572(73.2) | 338(79.7) | 172(72.4) | 112(79.4) | <0.001 |
| | YES | 2548(34.2) | 414(36.9) | 793(39.4) | 667(40.4) | 348(32.2) | 209(26.8) | 86(20.3) | 65(27.6) | 29(20.6) | |
| Race(%) | Mexican American | 678(9.1) | 185(16.5) | 219(10.9) | 163(9.9) | 81(7.5) | 35(4.5) | 21(4.9) | 11(4.6) | 5(3.2) | <0.001 |
| | Other Hispanic | 469(6.3) | 119(10.6) | 121(6) | 109(6.6) | 65(6) | 40(5.1) | 11(2.7) | 14(5.9) | 9(6.6) | |
| | Non-Hispanic White | 4901(65.8) | 487(43.4) | 1292(64.2) | 1076(65.2) | 742(68.6) | 588(75.3) | 324(76.3) | 176(74.1) | 108(76.6) | |
| | Non-Hispanic Black | 834(11.2) | 265(23.6) | 278(13.8) | 173(10.5) | 83(7.7) | 45(5.8) | 20(4.8) | 16(6.7) | 12(8.4) | |
| | Other Race—Including Multi-Racial | 574(7.7) | 66(5.9) | 105(5.2) | 127(7.7) | 110(10.2) | 73(9.3) | 48(11.4) | 21(8.7) | 7(5.2) | |
| Literacy(%) | Below high school | 1065(14.3) | 386(34.4) | 408(20.3) | 220(13.3) | 88(8.1) | 38(4.9) | 18(4.3) | 8(3.2) | 5(3.2) | <0.001 |
| | High school graduate | 1743(23.4) | 351(31.3) | 672(33.4) | 428(25.9) | 230(21.3) | 71(9.1) | 55(13) | 20(8.5) | 9(6.3) | |
| | High school or above | 4648(62.4) | 385(34.3) | 932(46.3) | 1004(60.8) | 763(70.6) | 672(86.1) | 350(82.6) | 209(88.3) | 128(90.5) | |
| Family PIR(%) | impoverished | 1669(22.4) | 450(40.1) | 638(31.7) | 357(21.6) | 199(18.4) | 77(9.9) | 33(7.7) | 20(8.4) | 13(9.1) | <0.001 |
| | medium | 2324(31.2) | 369(32.9) | 746(37.1) | 594(36) | 337(31.2) | 187(24) | 81(19) | 45(18.9) | 28(19.7) | |
| | wealthy | 2838(38.1) | 186(16.6) | 453(22.5) | 558(33.8) | 463(42.8) | 452(57.9) | 276(65.1) | 166(70) | 92(65.6) | |
| | Unknown | 618(8.3) | 117(10.4) | 175(8.7) | 142(8.6) | 83(7.7) | 63(8.1) | 35(8.2) | 6(2.7) | 8(5.6) | |
| Diabetes(%) | NO | 6674(89.6) | 981(87.4) | 1746(86.8) | 1453(88) | 989(91.5) | 721(92.3) | 404(95.3) | 222(93.6) | 130(92.4) | <0.001 |
| | YES | 775(10.4) | 141(12.6) | 266(13.2) | 198(12) | 92(8.5) | 60(7.7) | 20(4.7) | 15(6.4) | 11(7.6) | |
| Had a hysterectomy (%) | NO | 5669(76.1) | 799(71.2) | 1422(70.7) | 1159(70.2) | 868(80.3) | 636(81.4) | 382(90.1) | 209(88) | 124(88.1) | <0.001 |
| | YES | 1780(23.9) | 323(28.8) | 590(29.3) | 492(29.8) | 213(19.7) | 145(18.6) | 42(9.9) | 28(12) | 17(11.9) | |
| Both ovaries removed(%) | NO | 6503(87.3) | 987(88) | 1718(85.4) | 1364(82.6) | 964(89.2) | 701(89.8) | 398(93.8) | 219(92.3) | 130(91.9) | <0.001 |
| | YES | 946(12.7) | 135(12) | 294(14.6) | 287(17.4) | 117(10.8) | 80(10.2) | 26(6.2) | 18(7.7) | 11(8.1) | |
| Use female hormones (%) | NO | 5803(77.9) | 941(83.9) | 1551(77.1) | 1210(73.3) | 838(77.5) | 623(79.8) | 360(84.8) | 196(82.7) | 86(61.1) | <0.001 |
| | YES | 1646(22.1) | 181(16.1) | 461(22.9) | 441(26.7) | 243(22.5) | 158(20.2) | 64(15.2) | 41(17.3) | 55(38.9) | |
| Current smoker (%) | NO | 4648(62.4) | 579(51.6) | 1074(53.4) | 994(60.2) | 742(68.6) | 562(71.9) | 318(74.9) | 182(76.9) | 103(73.1) | <0.001 |
| | YES | 2801(37.6) | 544(48.5) | 938(46.6) | 657(39.8) | 339(31.4) | 219(28.1) | 106(25.1) | 55(23.1) | 38(26.9) | |
| Menopause(%) | NO | 3315(44.5) | 570(50.8) | 853(42.4) | 636(38.5) | 471(43.6) | 373(47.8) | 228(53.8) | 111(47) | 60(42.3) | <0.001 |
| | YES | 2577(34.6) | 303(27) | 650(32.3) | 614(37.2) | 422(39) | 263(33.7) | 138(32.6) | 100(42.3) | 60(42.5) | |
| | Unknown | 1557(20.9) | 249(22.2) | 509(25.3) | 401(24.3) | 188(17.4) | 144(18.5) | 58(13.6) | 25(10.7) | 21(15.2) | |
| Oral contraceptive (%) | NO | 1490(20) | 245(21.8) | 437(21.7) | 388(23.5) | 214(19.8) | 116(14.9) | 65(15.3) | 37(15.8) | 22(15.9) | 0.092 |
| | YES | 4745(63.7) | 703(62.7) | 1260(62.6) | 1032(62.5) | 697(64.5) | 522(66.9) | 271(63.9) | 157(66.3) | 89(63.1) | |
| | Unknown | 1214(16.3) | 174(15.5) | 314(15.6) | 231(14) | 170(15.7) | 142(18.2) | 88(20.8) | 42(17.9) | 30(21) | |

(*Continued*)

**Table 1.** (Continued)

| | | | Age at first birth (years) | | | | | | | | |
|---|---|---|---|---|---|---|---|---|---|---|---|
| | | Total (N = 7449) | <18 (N = 1122) | 18-20 (N = 2012) | 21-23 (N = 1651) | 24-26 (N = 1081) | 27-29 (N = 781) | 30-32 (N = 424) | 33-35 (N = 237) | >35 (N = 141) | P |
| RA(%) | NO | 7032(94.4) | 1014(90.4) | 1881(93.5) | 1549(93.8) | 1024(94.7) | 755(96.7) | 418(98.7) | 231(97.3) | 135(95.5) | <0.001 |
| | YES | 417(5.6) | 109(9.7) | 131(6.5) | 102(6.2) | 57(5.3) | 26(3.3) | 6(1.3) | 6(2.7) | 6(4.5) | |
| Age (year) | | 50.76±0.23 | 46.66±0.62 | 50.35±0.48 | 53.29±0.5 | 52.02±0.58 | 49.86 ±0.56 | 49.27±0.77 | 52.01±0.97 | 52.05 ±1.31 | <0.001 |
| TC (mmol/L) | | 5.07±0.02 | 4.99±0.04 | 4.97±0.03 | 5.08±0.04 | 5.11±0.04 | 5.11±0.05 | 5.14±0.07 | 5.24±0.08 | 5.34±0.1 | <0.001 |
| UA (μmol/L) | | 283.79±1.16 | 283.19 ±2.67 | 292.35±2.38 | 286.85±2.49 | 284.23±3.01 | 268.74 ±3.19 | 275.36 ±4.48 | 281.66 ±5.88 | 278.4±8.6 | 0.003 |

Continuous variables were shown as survey-weighted mean ± SE; Categorical variables expressed as No. (%).

Abbreviations:BMI body mass index,RA rheumatoid arthritis,TC total cholesterol,UA uric acid.

an AFB of 33–35 years relative to reference group (AFB of 30–32 years). After further adjustment for confounders, the association between an AFB > 35 years and the reference group (AFB of 30–32 years) became more significant. Even though after adjusting for sociodemographic data, reproductive factors, and lifestyle factors, the association was weakened but still present. We examined the nonlinear relation of AFB with RA and detected a notable trend (Fig 2).

With regard to the nonlinear association between AFB and RA, AFB showed the dose-response relation with RA risk. Model 3 was fully adjusted, and later restricted cubic spline analysis was conducted to model AFB. Notably, the solid and dashed lines stand for OR and 95% CI of the spline model.

There were variations between the predefined subgroups stratified by hypertension (yes/no), diabetes (yes/no), history of hysterectomy (yes/no), oophorectomy (yes/no), oral contraceptive use (yes/no), smoking status (yes/no), and alcohol consumption (yes/no). According to our results, AFB was more significantly associated with RA among hypertension, without diabetes, with hysterectomy, no smoking, and with alcohol consumption subgroups (Table 3). Most of the subgroups did not exhibit significant statistical interactions (P-interaction > 0.05). However, significant interactions were observed between AFB and hypertension (P = 0.028) and smoking (P = 0.042).

**Table 2. Survey weighted odds ratios (95% CI) for the association between age at first birth and the presence of rheumatoid arthritis.** From: Association between age at first birth and risk of rheumatoid arthritis in women: Evidence from NHANES 2011–2020.

| | Age at first birth (years) | | | | | | | |
|---|---|---|---|---|---|---|---|---|
| | <18 | 18–20 | 21–23 | 24–26 | 27–29 | 30–32 | 33–35 | >35 |
| Model1 | 8.04(3.63–17.79) p<0.001 | 5.26(2.41–11.5) p<0.001 | 4.94(2.26–10.79) p<0.001 | 4.19(1.84–9.57) p<0.001 | 2.59(1.1–6.11) p<0.030 | reference | 2.13(0.78–5.79) p<0.139 | 3.59(1.13–11.42) p<0.031 |
| Model2 | 7.01(3.12–15.76) p <0.001 | 4.49(2.04–9.89) p<0.001 | 3.95(1.8–8.65) p<0.001 | 3.59(1.57–8.2) p<0.002 | 2.51(1.06–5.94) p<0.037 | reference | 1.93(0.72–5.23) p<0.194 | 3.24(1.01–10.33) p<0.047 |
| Model3 | 4.01(1.71–9.4) p<0.001 | 2.91(1.25–6.73) p<0.013 | 3.01(1.33–6.8) p<0.008 | 3.19(1.37–7.43) p<0.007 | 2.39(0.99–5.76) p<0.052 | reference | 1.99(0.72–5.51) p<0.187 | 3.36(1.05–10.78) p<0.041 |

Model 1: Unadjusted.

Model 2: Adjusted for age (years) and race/ethnicity (Mexican American, other Hispanic, non-Hispanic White, non-Hispanic Black, other race).

Model 3: Adjusted for hypertension (yes/no), diabetes (yes/no), smoking status (yes/no), alcohol consumption (yes/no), menopausal status (yes/no), hormone therapy use (yes/no), oral contraceptive use (yes/no), history of hysterectomy (yes/no), bilateral oophorectomy (yes/no), education, family PIR, BMI, UA, and TC.

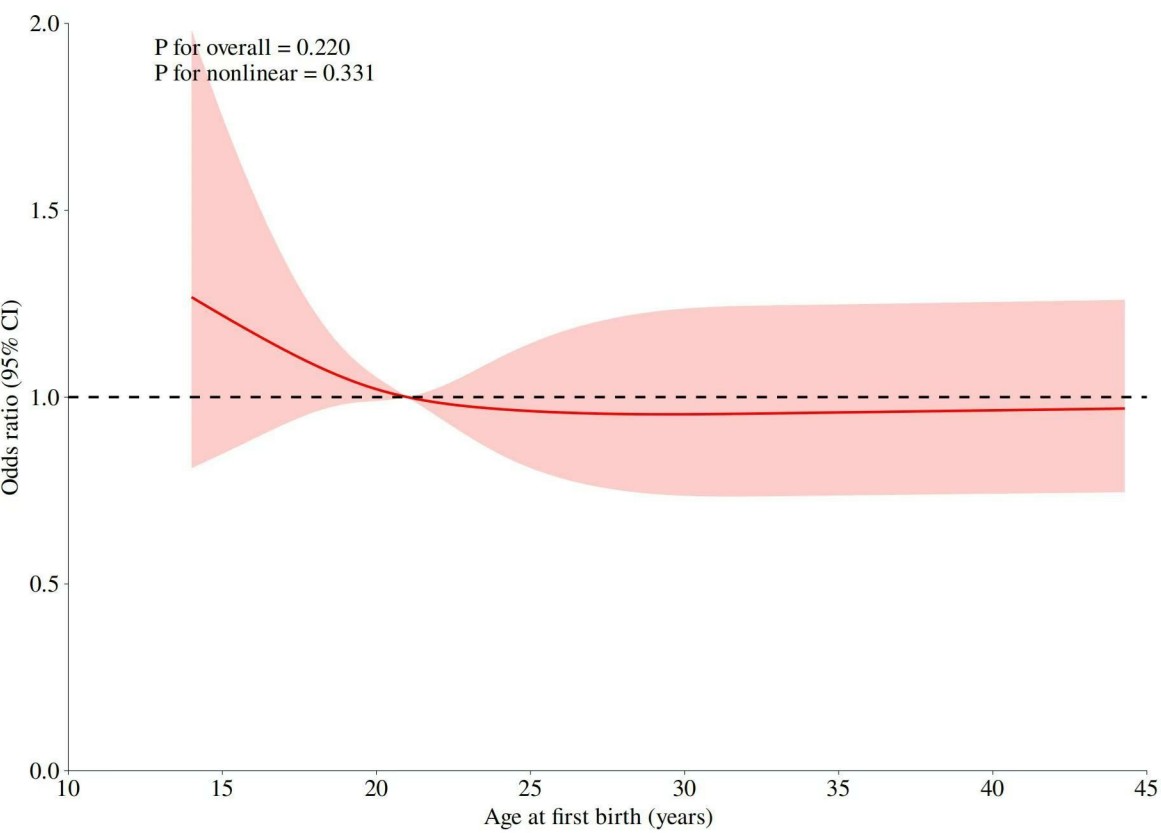

**Fig 2. From: Association between age at first birth and risk of rheumatoid arthritis in women: Evidence from NHANES 2011–2020.**

## Discussion

The present population-based study included a total of 7,449 women, among whom, 552 were diagnosed with RA. We found that women with an AFB < 26, or > 35 years had an increased RA risk in their later life, with the highest risk being observed in subjects having an AFB < 18 years. Subjects who had an AFB of 27–29 or 33–35 years were not significantly different from those in reference group (AFB of 30–32 years).

Previous studies have yielded inconsistent findings regarding the association between reproductive factors and the risk of rheumatoid arthritis (RA) in women. Zhu et al. conducted a two-sample Mendelian randomization (MR) study investigating the association between three relevant exposures—age at menarche (AAM), age at natural menopause (ANM), and age at first birth (AFB)—and the risk of rheumatoid arthritis (RA) [19]. However, they found no evidence to support a causal relationship between genetically predicted reproductive factors and RA risk. This discrepancy with our study results may be attributed to several factors: firstly, the relatively small sample size utilized by Zhu et al. may lead to insufficient statistical power and an increased risk of bias, which could undermine the persuasiveness of their conclusions. Secondly, AFB is a behavioral characteristic heavily influenced by socio-psychological, cultural, and economic factors rather than genetic factors, which their study did not adequately consider. Our study employed a complex, stratified, multistage, probability-cluster sampling method, considering multiple factors such as age, race, and socioeconomic status, covering a broad segment of the U.S. population and providing extensive data, ensuring the diversity and representativeness of the data.Hind A Beydoun et al. found that women who

**Table 3. Stratified analysis of the association between age at first birth and the presence of rheumatoid arthritis.** From: Association between age at first birth and risk of rheumatoid arthritis in women: Evidence from NHANES 2011–2020.

| | Age at first birth (years) | | | | | | | | |
|---|---|---|---|---|---|---|---|---|---|
| | <18 | 18–20 | 21–23 | 24–26 | 27–29 | 30–32 | 33–35 | >35 | P—interaction |
| **Hypertension** | | | | | | | | | **0.028** |
| NO | 2.87(0.90–9.18) | 2.58(0.80–8.35) | 2.79(0.96–8.11) | 2.13(0.69–6.57) | 1.77(0.57–5.47) | 1.00 (Ref.) | 1.99(0.56–7.14) | 2.6(0.53–12.73) | |
| YES | 7.59(2.20–26.16) | 4.64(1.38–15.60) | 4.76(1.40–16.24) | 6.08(1.70–21.68) | 4.16(1.06–16.40) | 1.00 (Ref.) | 2.04(0.45–9.25) | 5.49(1.31–23.11) | |
| Diabetes | | | | | | | | | 0.446 |
| NO | 4.69(1.71–12.84) | 3.2(1.18–8.73) | 3.26(1.24–8.56) | 3.75(1.39–10.11) | 2.56(0.93–7.06) | 1.00 (Ref.) | 1.96(0.58–6.68) | 4.09(1.08–15.54) | |
| YES | 1.57(0.41–5.98) | 1.64(0.46–5.87) | 1.92(0.53–7.01) | 1.29(0.33–5.00) | 1.63(0.30–8.86) | 1.00 (Ref.) | 1.58(0.31–8.15) | 1.12(0.16–7.83) | |
| Current smoker | | | | | | | | | 0.042 |
| NO | 4.04(1.67–9.77) | 4.54(1.95–10.59) | 4.36(1.94–9.80) | 3.56(1.50–8.42) | 2.31(0.92–5.79) | 1.00 (Ref.) | 2.79(0.97–8.04) | 2.77(0.97–7.93) | |
| YES | 3.59(0.66–19.62) | 1.73(0.32–9.46) | 1.84(0.35–9.73) | 2.90(0.51–16.41) | 2.46(0.42–14.27) | 1.00 (Ref.) | 0.78(0.10–6.17) | 4.38(0.49–39.6) | |
| Alcohol intake | | | | | | | | | 0.616 |
| NO | 1.73(0.57–5.25) | 0.82(0.27–2.46) | 1.03(0.37–2.90) | 0.81(0.27–2.43) | 0.37(0.10–1.36) | 1.00 (Ref.) | 0.49(0.10–2.52) | 0.52(0.08–3.42) | |
| YES | 3.78(1.13–12.66) | 2.54(0.77–8.37) | 3.33(1.06–10.5) | 2.45(0.70–8.56) | 2.28(0.66–7.88) | 1.00 (Ref.) | 2.20(0.52–9.22) | 2.34(0.64–8.57) | |
| Had a hysterectomy | | | | | | | | | 0.084 |
| NO | 3.79(1.44–9.99) | 2.58(0.99–6.74) | 2.71(1.08–6.79) | 2.63(1.02–6.77) | 2.28(0.89–5.87) | 1.00 (Ref.) | 1.3(0.46–3.69) | 1.64(0.54–4.96) | |
| YES | 8.52(1.49–48.71) | 6.64(1.21–36.5) | 6.46(1.19–35.22) | 8.65(1.48–50.6) | 3.74(0.48–28.87) | 1.00 (Ref.) | 8.75(0.81–94) | 15.93(1.35–187.81) | |
| Both ovaries removed | | | | | | | | | 0.057 |
| NO | 3.77(1.46–9.73) | 2.58(1.02–6.52) | 2.87(1.17–7.05) | 3.16(1.25–7.98) | 2.14(0.84–5.48) | 1.00 (Ref.) | 1.44(0.52–3.98) | 1.67(0.55–5.02) | |
| YES | 11.7(1.02–134.75) | 9.09(0.89–92.64) | 6.77(0.63–72.22) | 5.96(0.53–66.88) | 5.29(0.35–79.21) | 1.00 (Ref.) | 14.47(0.5–420.68) | 39.89(2.43–655.89) | |
| Oral contraceptive | | | | | | | | | 0.787 |
| NO | 5.35(1.3–22.05) | 2.29(0.54–9.67) | 3.28(0.83–12.85) | 2.59(0.59–11.38) | 1.27(0.25–6.5) | 1.00 (Ref.) | 2.49(0.45–13.69) | 2.96(0.54–16.29) | |
| YES | 3.23(1.01–10.3) | 2.57(0.83–7.99) | 2.46(0.81–7.48) | 3.17(1.02–9.88) | 2.34(0.73–7.53) | 1.00 (Ref.) | 1.98(0.49–8) | 2.67(0.43–16.73) | |

Adjusted for model 3.

experience menopause before the age of 40 seem to be at an increased risk of developing post-menopausal rheumatoid arthritis, and that age at menarche and pregnancy history may not predict postmenopausal RA [20]. They failed to find statistically significant relationships between postmenopausal RA and other characteristics (including age at menarche, pregnancy history, number of pregnancies, number of live births, age at first live birth, age at last live birth, hysterectomy status, oophorectomy status, and the use of oral contraceptives or hormone therapy). However, their study could only verify RA diagnoses in women aged 60 or older, limiting the generalizability of their findings. Our study also utilized the NHANES

database for analysis but targeted a wider age range of women, making our findings more widely applicable and a valuable supplement to previous related studies.

Cecilia Orellana et al. conducted a case-control study suggesting that the increased risk of RA in women seems to be associated with increased risk postpartum and a younger AFB [21]. They believed that women who give birth to their first child before the age of 23 have a significantly increased chance of developing RA. A prospective cohort study in France also found that an earlier AFB, specifically under 22 years of age, is associated with an increased risk of RA [22]. Other relevant studies have also arrived at the same conclusion that an earlier AFB increases the risk of developing RA [23]. However, they did not specify the underlying reasons. This is somewhat similar to our study's findings, which suggest that women with AFB < 26 years are more likely to develop RA. Unlike their study, our research found that women with AFB > 35 years also exhibit an increased risk of RA. Further biological mechanism studies are needed to explain this phenomenon.

The conclusion drawn from this study is that Women with an AFB < 26, or > 35 years have a higher risk of developing RA later in life.We hypothesize that the increased risk of RA in women who give birth at a younger age may be associated with changes in the immune system and hormonal levels during early pregnancy, which could affect long-term immune regulation. Previous studies have found that RA symptoms often alleviate during pregnancy and lactation, which is believed to be due to changes in hormone levels, particularly the elevation of estrogen and progesterone [24–26]. If the AFB is early, this protective effect may dissipate sooner, thereby increasing the risk of developing RA.Women who have their first child at an older age may be more susceptible to rheumatoid arthritis (RA) due to the decline in ovarian function and changes in hormonal levels, which could affect the immune system and thereby increase the risk of RA [27–29]. Secondly, late childbearing may imply that individuals have a longer period to accumulate exposures to environmental and lifestyle factors prior to giving birth, which could potentially increase the risk of developing RA [30,31].

In addition to AFB, this study also included reproductive factors such as menopausal status, hormone use, history of hysterectomy or oophorectomy, and the use of oral contraceptives. Our subgroup analysis also found that women undergoing hysterectomy or oophorectomy had an increased risk of RA, consistent with previous findings [32]. According to our results, hysterectomy and oophorectomy (medically induced menopause) both increased the risk of RA. Additionally, we found little correlation between oral contraceptive use and RA risk. Previous research has shown no significant impact of oral contraceptive use on the RA risk. Such findings conform to the meta-analysis of cohort studies, which demonstrates that oral contraceptive use duration does not show dose-response relationship with RA risk [33].

In addition to physiological reasons, the results of this study can also be explained from a socioeconomic perspective. For instance, women who give birth at a younger age may have the lower education level and family PIR. These factors are often associated with unhealthy behaviors, such as smoking, poor physical fitness, and high alcohol consumption, all of which may result in a higher RA risk later in life. Additionally, women with a younger AFB are more likely to experience unintended pregnancies, which may be linked to unhealthy behaviors during pregnancy or delayed access to healthcare.

The present work shows multiple strengths. Firstly, this is one of the rare studies to examine the relationship between AFB and RA. Our findings provide robust insights into the RA risk factors and associated diseases. Secondly, this study was conducted using NHANES data, which utilizes the population-based design and has a large sample size, with racial diversity, and a well-characterized study population. The database provides more details about reproductive hormone-related factors as well as additional key covariates, and its large sample size supports stratified analyses by age and race.

However, there are several limitations in this study. The findings of this study exhibit a high degree of generalizability within the American population; however, due to cultural and genetic disparities, they may not be directly applicable to other populations. A significant portion of the data, like RA history and reproductive health, were self-reported, which probably caused recall bias and reduced our result reliability. Besides, due to the cross-sectional design, it was impossible to establish clear causal relationships. Finally, our data were drawn in several cross-sectional studies, which suggested the non-uniform questionnaire design among all survey cycles, potentially leading to unmeasured biases during data integration. Therefore, further investigations are warranted for validating our results.

To sum up, women who have an AFB < 26, or > 35 years are associated with an increased RA risk in their later life. Policymakers may consider focusing more on women in these AFB age ranges in screening RA and making preventive measures.

## Acknowledgments

The authors acknowledge the participants and staff of the National Health and Nutrition Examination Survey (NHANES) for their dedication and contribution to the research.

## Author Contributions

**Data curation:** Chenyang Huang.

**Formal analysis:** Chenyang Huang.

**Investigation:** Tianci Guo.

**Methodology:** Yuntian Ye.

**Resources:** Weijie Yu.

**Software:** Puyu Niu.

**Supervision:** Kuiliang Gao.

**Validation:** Jiajin Liu.

**Writing – original draft:** Hang Cui.

**Writing – review & editing:** Aifeng Liu.

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
