## [Decision Letter · Decision Letter 0]

6 Dec 2024

PONE-D-24-53513Association between age at first birth and risk of rheumatoid arthritis in women: evidence from NHANES 2011–2020PLOS ONE

Dear Dr. Liu,

Thank you for submitting your manuscript to PLOS ONE. After careful consideration, we feel that it has merit but does not fully meet PLOS ONE’s publication criteria as it currently stands. Therefore, we invite you to submit a revised version of the manuscript that addresses the points raised during the review process.

We look forward to receiving your revised manuscript.

Kind regards,

Ahmed Qasim Mohammed Alhatemi

Academic Editor

PLOS ONE

Journal Requirements:

2. Thank you for stating the following financial disclosure: Funding

1.National Natural Science Foundation of China General Program (82474538);2.Tianjin Municipal Health Commission Jinmen Medical Talents Program (TJSJMYXYC-D2-028);3.Tianjin Science and Technology Plan Project (23KPXMRC00170).

Reviewers' comments:

Reviewer's Responses to Questions

**Comments to the Author**

1. Is the manuscript technically sound, and do the data support the conclusions?

Reviewer #1: No

Reviewer #2: Yes

Reviewer #3: Partly

2. Has the statistical analysis been performed appropriately and rigorously? 

Reviewer #1: I Don't Know

Reviewer #2: Yes

Reviewer #3: No

3. Have the authors made all data underlying the findings in their manuscript fully available?

Reviewer #1: Yes

Reviewer #2: Yes

Reviewer #3: Yes

4. Is the manuscript presented in an intelligible fashion and written in standard English?

Reviewer #1: Yes

Reviewer #2: Yes

Reviewer #3: No

5. Review Comments to the Author

Reviewer #1: The results of the study do not go in line with what we already know about the subject. Discussion is poorly written and needs major revisions. I am also highlighting some other queries

Major comment 1

Why wasn’t a case-control approach taken for statistical analysis in this cross-sectional study? Taking a particular age group as reference and comparing others with it seems like an unorthodox approach.

Major comment 2

The whole discussion needs to be revised. Overall, it feels that most of the content written in discussion is meant to fill pages and has little relevance with the results of the study at hand.

There is a lot of repetition related to background of RA in discussion in 2nd paragraph. Please remove that.

The pathogenesis related to interaction of RA with hormonal and fertility factors has been described well but it is very extensive. It covers more than half of discussion and should be shortened. Discussion is meant to discuss the results and not the etiopathogenesis. Ideally the pathogenesis or concept behind the study premise is discussed in introduction.

The author has used the line in discussion “Firstly, this study is among those several articles examining the relationship of AFB with RA”. There are several studies in the literature which show that earlier AFB is associated with a lower risk for RA, contrary to the results of this study. Please describe these studies and compare your results with them.

Minor comment 1

In discussion, the line “Among women with an AFB of 18-32 years, the RA prevalence showed a gradual downward trend, while women with an AFB > 35 years exhibited an upward trend in terms of the RA prevalence” is contradictory to the authors results. Please correct it.

Minor Comment 2

In the abstract and conclusion, instead of mentioning all the different age categories separately (AFB < 18, 18–20, 21–23, 24–26,), it would be better to write as 18-26 years of age group.

Minor comment 3

In the second paragraph of introduction “Women's health is influenced by various reproductive factors, which may be important for OA occurrence”, correct OA to RA

Minor comment 4

In the abstract, in methods section, the author can’t write that data has been taken from women >20 years of age since the study population contains women less than 18 years of age also. Please correct it.

Reviewer #2: In the article, the age at first birth for women aged 27-35 did not show a significant difference in terms of rheumatoid arthritis (RA) risk compared to other age groups. In other words, the first birth in this age range did not significantly increase the risk of RA. However, these results are based on analyses conducted on the NHANES database, and caution should be exercised when making generalizations. Because RA risk can vary depending on many factors, and such epidemiological studies only show associations, not cause-and-effect relationships.

The study was conducted using the NHANES database, which provides a large and diverse sample. Analyses were conducted using logistic regression models, which is an appropriate method because it allows for the evaluation of relationships and probabilities among various variables. However, since the study had a cross-sectional design, it may be difficult to establish cause-and-effect relationships. The study also made detailed adjustments by controlling for age, race, education level, and many other potential variables. The NHANES database includes individuals from various ethnic and socioeconomic groups across the United States, which makes the findings highly generalizable to the U.S. population.

However, the effects may not be the same across populations in different countries. The scope of the study focused on a specific factor, namely childbearing history, which may affect the risk of RA in women. There are limited studies in the literature on childbearing history and RA risk. The results of this study may be consistent with the findings of previous studies, but more studies are needed in this area. In particular, there is evidence that giving birth at an earlier age may increase the risk of RA, and giving birth at a later age may also pose a potential risk. The article stated that it has no financial conflict of interest.

Critiques:

Data Source and Methodology: The use of the NHANES database provides a diverse and extensive sample; however, the cross-sectional study design limits the ability to establish cause-and-effect relationships. Self-reported data may introduce recall bias affecting the study's reliability.

Analysis Methods: While logistic regression is suitable for examining relationships among variables, details on model adequacy and validation should be further addressed.

Generalizability: Findings are highly generalizable within the U.S. population but may not directly apply to other populations due to cultural and genetic differences.

Consistency with Literature: The findings align with previous studies; however, this area of research has limited literature, highlighting the need for further studies to confirm these observations.

Suggestions:

Multi-Center and Longitudinal Studies: To better understand the etiology of RA, conducting multi-center and longitudinal studies across different populations would be beneficial.

Detailed Hormonal Profile Studies: Exploring the effects of other hormones such as FSH and prolactin on RA could provide deeper insights into the hormonal influences on autoimmune diseases.

Socioeconomic Factors Analysis: Investigating the impact of socioeconomic factors on RA risk could elucidate additional determinants that affect health outcomes.

Public Health and Education Programs: Based on the findings, targeting women in high-risk age groups for RA screening and prevention should be considered to mitigate disease impact.

Conclusion: The manuscript provides valuable insights into the epidemiological factors associated with RA in women. Enhancements in data analysis robustness and a broader consideration of socioeconomic and hormonal variables could further strengthen the conclusions drawn.

Reviewer #3: The manuscript provides valuable insights into the relationship between reproductive history and rheumatoid arthritis. However, several aspects of the methodology, organization, and presentation could be improved to enhance the study’s clarity, rigor, and contribution to the field. Below are specific comments and suggestions for improvement:

1. Addressing Multicollinearity

Motivational Context:

In multivariable regression models, multicollinearity between independent variables can significantly affect the stability and interpretability of the estimated coefficients. Given the comprehensive set of covariates included in the fully adjusted model (e.g., age, BMI, hypertension, diabetes, and others), addressing potential multicollinearity is crucial to ensure reliable results and valid conclusions.

Question to the Authors:

Could the authors clarify how multicollinearity was assessed and managed during the development of the regression models? Specifically, were any diagnostic measures (e.g., Variance Inflation Factor, correlation matrices) employed, and were any corrective actions, such as variable selection or regularization techniques, undertaken to mitigate its effects?

2. Use of STATA for Statistical Analysis

The use of STATA version 18.0 for statistical analyses is a valid and robust choice, particularly given the complexities of the NHANES design and the application of Mobile Examination Center (MEC) sampling weights. However, the current description in the manuscript ("due to the complicated NHANES design") is somewhat generalized and could benefit from additional specificity.

For instance, it would be helpful to explicitly state which aspects of the NHANES design (e.g., stratification, multistage sampling, clustering) and the need to apply sampling weights influenced the decision to use STATA. This would not only clarify the rationale behind the methodological choice but also enhance the scientific rigor and transparency of the article.

3. Formatting and Organization of the "Methods Statements" Section

a) Section Title:

The title of the section "Methods Statements" is written in lowercase and bold font, which deviates from the formatting style used in the other subsections of the "Methods" section, where titles are capitalized and not bolded. To ensure consistency and professionalism, we recommend aligning this title with the formatting of other subsections.

b) Additionally, the term "Methods Statements" might not be the most appropriate for this context. Considering the content and structure of the article, the "Methods" section could be more aptly titled "Materials and Methods," as it includes a description of "Study Subjects," which represents the material of the research.

c) Duplication of Content:

The statements included in the "Methods Statements" subsection are repeated in the subsequent "Study Subjects" subsection. This redundancy does not add value and could be streamlined by removing the duplication or integrating the information into the appropriate subsection.

d) Highlighted Text:

A fragment of the text in the "Methods Statements" subsection is highlighted in red. While such formatting might have been used during the drafting or editing process, it appears inappropriate in a finalized manuscript. We recommend removing the red highlight to ensure consistency with standard scientific formatting and improve the professional presentation of the article.

4. Contextualization with Prior Research

While the current study provides valuable insights into the relationship between reproductive history and rheumatoid arthritis, it appears that the article does not reference prior research addressing similar topics using the same dataset.

For example, the study "Reproductive history and postmenopausal rheumatoid arthritis among women 60 years or older: Third National Health and Nutrition Examination Survey" (DOI: 10.1097/gme.0b013e3182a14372), published in 2013, explores the association between reproductive factors and RA, including age at first pregnancy, using data from NHANES, the same dataset employed in the present study.

Inclusion of such prior research in the discussion section would provide a more comprehensive contextualization of the current findings. Specifically, comparing the methodologies, population characteristics, and key results of the present study with those of earlier works could clarify how the current research builds upon, diverges from, or complements existing knowledge derived from the same dataset.

5. Conclusion Section

In the conclusion, the statement reads:

“Women with an AFB < 18, 18–20, 21–23, 24–26, or > 35 years have a higher risk of developing RA later in life. Policymakers must focus more on women in these AFB age ranges in screening RA and making preventive measures.”

This conclusion would be clearer and more impactful if it summarized the risk for broader age categories instead of listing specific intervals. For example, grouping women as "younger than 26 years" rather than specifying "<18, 18–20, 21–23, 24–26" would make the statement more concise and generalizable.

6. Inconsistencies in References

The formatting of the references in the manuscript is inconsistent:

a) Author names are written in a mix of uppercase and lowercase letters.

Spacing is inconsistent after punctuation marks.

b) Page ranges are inconsistently formatted, alternating between short dashes (e.g., 73–7) and long dashes with spaces (e.g., 3218 － 3228).

It is recommended to thoroughly review the reference list and ensuring uniform formatting that adheres to the chosen citation style. Consistency in formatting is essential for maintaining the professionalism and credibility of the manuscript.

By addressing these points, the authors can enhance the clarity, rigor, and overall impact of their study.

6. PLOS authors have the option to publish the peer review history of their article (what does this mean?). If published, this will include your full peer review and any attached files.

Reviewer #1: No

Reviewer #2: **Yes: **Selçuk Yüksel

Reviewer #3: **Yes: **Artyom Borbat

---

## [Author Response · Author response to Decision Letter 0]

15 Dec 2024

[Aifeng Liu]

[First Teaching Hospital of Tianjin University of Traditional Chinese Medicine, Tianjin 300381, China]

[draifeng@163.com]

Editor-in-Chief,

PLOS ONE

Subject: Response to Reviewer Comments - Manuscript ID [PONE-D-24-53513]

Dear Editor-in-Chief,

We sincerely appreciate the valuable feedback provided by both the reviewers and the Editor for our manuscript titled " Association between age at first birth and risk of rheumatoid arthritis in women:evidence from NHANES 2011–2020" (Manuscript ID: PONE-D-24-53513). We have thoroughly reviewed and carefully considered all the comments received, and we have made comprehensive revisions to the manuscript, with text changes highlighted in red, to enhance its quality and clarity. Please find below our point-by-point response to the comments received:

**Reviewer 1:**

[Major comment 1] • Why wasn’t a case-control approach taken for statistical analysis in this cross-sectional study? Taking a particular age group as reference and comparing others with it seems like an unorthodox approach.

Response: Thank you for your insightful comment regarding the study design of the NHANES (National Health and Nutrition Examination Survey). We appreciate your attention to the methodological details. NHANES is a cross-sectional study, which does not differentiate between case and control groups but is based on sampling from the entire population, thus making it incapable of conducting direct case-control analyses.

Below are the references for the research methods we consulted. The study uses a specific age group as a reference and compares other age groups to it:

1.Yang HH, Chen GC, Zhou MG, Xie LF, Jin YY, Chen HT, Chen ZK, Kong YH, Yuan CZ, Li ZH. Association of age at first birth and risk of non-alcoholic fatty liver disease in women: evidence from the NHANES. Hepatol Int. 2023 Apr;17(2):303-312. doi: 10.1007/s12072-022-10429-1. Epub 2022 Oct 13. PMID: 36227515.

[Major comment 2] •  The whole discussion needs to be revised. Overall, it feels that most of the content written in discussion is meant to fill pages and has little relevance with the results of the study at hand.

There is a lot of repetition related to background of RA in discussion in 2nd paragraph. Please remove that.

The pathogenesis related to interaction of RA with hormonal and fertility factors has been described well but it is very extensive. It covers more than half of discussion and should be shortened. Discussion is meant to discuss the results and not the etiopathogenesis. Ideally the pathogenesis or concept behind the study premise is discussed in introduction.

The author has used the line in discussion “Firstly, this study is among those several articles examining the relationship of AFB with RA”. There are several studies in the literature which show that earlier AFB is associated with a lower risk for RA, contrary to the results of this study. Please describe these studies and compare your results with them.

Response:Thank you for your suggestion. You raise a great point; We have made significant revisions to the discussion section of the article; please refer to the discussion section for the specific content.（Please refer to lines 136-188 in the manuscript.）

[Minor comment 1] •In discussion, the line “Among women with an AFB of 18-32 years, the RA prevalence showed a gradual downward trend, while women with an AFB > 35 years exhibited an upward trend in terms of the RA prevalence” is contradictory to the authors results. Please correct it.

Response: Thank you for your feedback. The statement "Among women with an AFB of 18-32 years, the RA prevalence showed a gradual downward trend, while women with an AFB > 35 years exhibited an upward trend in terms of the RA prevalence" has been deleted.

[Minor Comment 2] • In the abstract and conclusion, instead of mentioning all the different age categories separately (AFB < 18, 18–20, 21–23, 24–26,), it would be better to write as 18-26 years of age group.

Response: Thank you for your inquiry. We have summarized the different age categories.（Please refer to lines 34 and 187 in the manuscript.）

[Minor comment 3] • In the second paragraph of introduction “Women's health is influenced by various reproductive factors, which may be important for OA occurrence”, correct OA to RA 

Response: Thank you for your suggestion. We have replaced "OA" with "RA" in the second paragraph of the introduction.（Please refer to line 45 in the manuscript.）

[Minor comment 4] •In the abstract, in methods section, the author can’t write that data has been taken from women >20 years of age since the study population contains women less than 18 years of age also. Please correct it.

Response: Thank you for your suggestion. We have already revised the age restrictions in the methodology section of the abstract.（Please refer to line 26 in the manuscript.）

**Reviewer 2:**

[Comment 1] •In the article, the age at first birth for women aged 27-35 did not show a significant difference in terms of rheumatoid arthritis (RA) risk compared to other age groups. In other words, the first birth in this age range did not significantly increase the risk of RA. However, these results are based on analyses conducted on the NHANES database, and caution should be exercised when making generalizations. Because RA risk can vary depending on many factors, and such epidemiological studies only show associations, not cause-and-effect relationships.

The study was conducted using the NHANES database, which provides a large and diverse sample. Analyses were conducted using logistic regression models, which is an appropriate method because it allows for the evaluation of relationships and probabilities among various variables. However, since the study had a cross-sectional design, it may be difficult to establish cause-and-effect relationships. The study also made detailed adjustments by controlling for age, race, education level, and many other potential variables. The NHANES database includes individuals from various ethnic and socioeconomic groups across the United States, which makes the findings highly generalizable to the U.S. population.

However, the effects may not be the same across populations in different countries. The scope of the study focused on a specific factor, namely childbearing history, which may affect the risk of RA in women. There are limited studies in the literature on childbearing history and RA risk. The results of this study may be consistent with the findings of previous studies, but more studies are needed in this area. In particular, there is evidence that giving birth at an earlier age may increase the risk of RA, and giving birth at a later age may also pose a potential risk. The article stated that it has no financial conflict of interest.

Critiques:

Data Source and Methodology: The use of the NHANES database provides a diverse and extensive sample; however, the cross-sectional study design limits the ability to establish cause-and-effect relationships. Self-reported data may introduce recall bias affecting the study's reliability.

Analysis Methods: While logistic regression is suitable for examining relationships among variables, details on model adequacy and validation should be further addressed.

Generalizability: Findings are highly generalizable within the U.S. population but may not directly apply to other populations due to cultural and genetic differences.

Consistency with Literature: The findings align with previous studies; however, this area of research has limited literature, highlighting the need for further studies to confirm these observations.

Suggestions:

Multi-Center and Longitudinal Studies: To better understand the etiology of RA, conducting multi-center and longitudinal studies across different populations would be beneficial.

Detailed Hormonal Profile Studies: Exploring the effects of other hormones such as FSH and prolactin on RA could provide deeper insights into the hormonal influences on autoimmune diseases.

Socioeconomic Factors Analysis: Investigating the impact of socioeconomic factors on RA risk could elucidate additional determinants that affect health outcomes.

Public Health and Education Programs: Based on the findings, targeting women in high-risk age groups for RA screening and prevention should be considered to mitigate disease impact.

Conclusion: The manuscript provides valuable insights into the epidemiological factors associated with RA in women. Enhancements in data analysis robustness and a broader consideration of socioeconomic and hormonal variables could further strengthen the conclusions drawn.

Response:Thank you for your feedback. 

First and foremost, I would like to extend my deepest gratitude for the insightful comments and valuable suggestions you have provided. Your expert critique is of paramount importance to our research endeavors.

Regarding the Data Source and Methodology: We fully concur with your assessment that the NHANES database offers a diverse and extensive sample set, which is a significant advantage of our study. Concurrently, we acknowledge the limitations of a cross-sectional study design in establishing causality. We have further emphasized this limitation in the discussion section and highlighted the potential recall bias introduced by self-reported data. We have augmented our discussion on the reliability of the data and are considering the adoption of a prospective design in future studies to mitigate these potential biases.

Regarding the Analysis Methods: Your evaluation of the applicability of the logistic regression model is highly pertinent. We have detailed the selection and validation processes of the model in the methods section to ensure the transparency and adequacy of our analytical approach.

Regarding Generalizability: We recognize that, despite the NHANES database encompassing individuals from various ethnicities and socioeconomic groups in the United States, our findings may not be entirely generalizable to populations in other countries or regions. We have clearly stated this in the discussion section and emphasized the need for further research in diverse populations to validate our findings.

Regarding Consistency with Literature: We appreciate your observation that our research findings are consistent with existing literature. We agree that the literature in this field is limited and requires additional studies to support our observations. We have expanded our discussion in the literature review section to include this perspective and underscore the contributions of our study and future research directions.

Regarding Your Suggestions: We take your suggestions very seriously and plan to take the following actions:

1.Explore the possibility of conducting multicenter and longitudinal studies to better understand the etiology of RA.

2.Consider undertaking detailed hormonal profiling studies to gain a deeper understanding of the impact of hormones on RA.

3.Investigate the impact of socioeconomic factors on RA risk to reveal other determinants affecting health outcomes.

4.Based on our study's results, consider implementing RA screening and prevention programs targeted at high-risk age groups of women.

We appreciate your evaluation of the manuscript, which provides valuable insights into the epidemiological factors associated with RA in women. We have endeavored to enhance the robustness of our data analysis and to more broadly consider socioeconomic factors, thereby further strengthening our conclusions.

We are grateful for your meticulous review and valuable suggestions. We have incorporated these recommendations into our research to elevate the quality of our work.

Thank you once again for your detailed examination and insightful advice.

**Reviewer 3:**

[Comment 1] •1. Addressing Multicollinearity

Motivational Context:

In multivariable regression models, multicollinearity between independent variables can significantly affect the stability and interpretability of the estimated coefficients. Given the comprehensive set of covariates included in the fully adjusted model (e.g., age, BMI, hypertension, diabetes, and others), addressing potential multicollinearity is crucial to ensure reliable results and valid conclusions.

Question to the Authors:

Could the authors clarify how multicollinearity was assessed and managed during the development of the regression models? Specifically, were any diagnostic measures (e.g., Variance Inflation Factor, correlation matrices) employed, and were any corrective actions, such as variable selection or regularization techniques, undertaken to mitigate its effects?

Response: Thank you for your in-depth feedback on our study. In the process of model construction, we conducted multiple screenings through literature research, theoretical support, and data characteristics to ensure that the covariates included in the model are independent and significantly contribute to the research conclusions. During the screening process, we paid particular attention to the correlation between variables, avoiding the inclusion of highly correlated variables in the model simultaneously.

[Comment 2] •2. Use of STATA for Statistical Analysis

The use of STATA version 18.0 for statistical analyses is a valid and robust choice, particularly given the complexities of the NHANES design and the application of Mobile Examination Center (MEC) sampling weights. However, the current description in the manuscript ("due to the complicated NHANES design") is somewhat generalized and could benefit from additional specificity.

For instance, it would be helpful to explicitly state which aspects of the NHANES design (e.g., stratification, multistage sampling, clustering) and the need to apply sampling weights influenced the decision to use STATA. This would not only clarify the rationale behind the methodological choice but also enhance the scientific rigor and transparency of the article.

Response: Thank you for your valuable feedback on the article. In the Materials and Methods section of our article, we have provided additional information:The NHANES is a complex, stratified, multistage, probability-cluster program designed to assess Americans’ health and nutritional status. （Please refer to line 57 in the manuscript.）

[Comment 3] •3. Formatting and Organization of the "Methods Statements" Section

a) Section Title:

The title of the section "Methods Statements" is written in lowercase and bold font, which deviates from the formatting style used in the other subsections of the "Methods" section, where titles are capitalized and not bolded. To ensure consistency and professionalism, we recommend aligning this title with the formatting of other subsections.

b) Additionally, the term "Methods Statements" might not be the most appropriate for this context. Considering the content and structure of the article, the "Methods" section could be more aptly titled "Materials and Methods," as it includes a description of "Study Subjects," which represents the material of the research.

c) Duplication of Content:

The statements included in the "Methods Statements" subsection are repeated in the subsequent "Study Subjects" subsection. This redundancy does not add value and could be streamlined by removing the duplication or integrating the information into the appropriate subsection.

d) Highlighted Text:

A fragment of the text in the "Methods Statements" subsection is highlighted in red. While such formatting might have been used during the drafting or editing process, it appears inappropriate in a finalized manuscript. We recommend removing the red highlight to ensure consistency with standard scientific formatting and improve the professional presentation of the article.

Response: Thank you for your valuable feedback on the content and formatting of the article.

a)We have standardized the formatting of the Methods Statements section.

b)We have revised the section previously titled 'Methods' to 'Materials and Methods' to better reflect the comprehensive nature of our experimental procedures and the importance of the materials used.

c)We have streamlined and summarized the content of the Methods Statements section.

d)We have standardized the font color throughout the Methods 

---

## [Decision Letter · Decision Letter 1]

22 Dec 2024

PONE-D-24-53513R1Association between age at first birth and risk of rheumatoid arthritis in women: evidence from NHANES 2011–2020PLOS ONE

Dear Dr. Liu,

Thank you for submitting your manuscript to PLOS ONE. After careful consideration, we feel that it has merit but does not fully meet PLOS ONE’s publication criteria as it currently stands. Therefore, we invite you to submit a revised version of the manuscript that addresses the points raised during the review process. 

We look forward to receiving your revised manuscript.

Kind regards,

Ahmed Qasim Mohammed Alhatemi

Academic Editor

PLOS ONE

Journal Requirements:

Reviewers' comments:

Reviewer's Responses to Questions

**Comments to the Author**

1. If the authors have adequately addressed your comments raised in a previous round of review and you feel that this manuscript is now acceptable for publication, you may indicate that here to bypass the “Comments to the Author” section, enter your conflict of interest statement in the “Confidential to Editor” section, and submit your "Accept" recommendation.

Reviewer #1: All comments have been addressed

Reviewer #2: All comments have been addressed

Reviewer #3: (No Response)

2. Is the manuscript technically sound, and do the data support the conclusions?

Reviewer #1: Yes

Reviewer #2: Yes

Reviewer #3: Partly

3. Has the statistical analysis been performed appropriately and rigorously? 

Reviewer #1: Yes

Reviewer #2: Yes

Reviewer #3: Yes

4. Have the authors made all data underlying the findings in their manuscript fully available?

Reviewer #1: Yes

Reviewer #2: Yes

Reviewer #3: Yes

5. Is the manuscript presented in an intelligible fashion and written in standard English?

Reviewer #1: Yes

Reviewer #2: Yes

Reviewer #3: Yes

6. Review Comments to the Author

Reviewer #1: Thank you for making the suggested corrections. All the best for your manuscript

Minor comment 1

Please correct the abstract on the title page of the manuscript also.

Reviewer #2: 24 / 5.000

I found the revision sufficient.The authors have positively evaluated my suggested contributions in their articles. The changes made are satisfactory to me.

Reviewer #3: The revised manuscript demonstrates significant improvements in clarity, particularly in the description of statistical methods. The question raised in the initial review regarding the selection of factors for multivariable analysis is now resolved, as the authors have provided a clearer explanation of their analytical approach. This enhances the transparency and robustness of the study.

However, some aspects of the manuscript still warrant further attention:

1. Inconsistency in Claims about Previous Research

The introduction claims that no studies have specifically examined the relationship between Age at First Birth (AFB) and rheumatoid arthritis (RA) risk. However, the discussion section references several studies that indirectly or directly address this topic. This discrepancy undermines the novelty of the study and should be clarified.

2. Stylistic Deviations and Non-Academic Tone

The manuscript contains several instances of stylistic deviations from an academic tone, such as:

a) The phrase, "Policymakers must focus more on women in these AFB age ranges in screening RA and making preventive measures," employs emotionally charged language ("must") and shifts the tone from scientific to political. A more neutral phrasing is recommended.

b) A subjective critique of Zhu et al.'s study, emphasizing insufficient sample size without detailed justification.

c) Statements regarding the increased risk due to late childbirth ("longer period to accumulate exposures to environmental and lifestyle factors") presented as factual without sufficient evidence.

d) These examples are not exhaustive; the authors are encouraged to thoroughly review the manuscript for similar stylistic deviations to enhance its academic rigor.

3. Lack of Control for Multiple Comparisons and Type I Errors

The manuscript reports multiple statistically significant findings (P < 0.05) across various subgroups and categories but does not address whether adjustments for multiple comparisons were applied. This raises concerns about inflated Type I error rates (false positives). The absence of correction methods like Bonferroni or Holm adjustments undermines the validity of some conclusions, particularly given the extensive number of statistical tests performed. The authors should clarify whether such adjustments were applied or acknowledge this as a limitation in the discussion.

In conclusion, the manuscript provides valuable insights into the relationship between AFB and RA risk, and the improvements made after the initial review are commendable. Addressing the points mentioned above will further enhance the academic rigor and overall quality of the study.

7. PLOS authors have the option to publish the peer review history of their article (what does this mean?). If published, this will include your full peer review and any attached files.

Reviewer #1: No

Reviewer #2: **Yes: **Selçuk Yüksel

Reviewer #3: **Yes: **Artyom Borbat

---

## [Author Response · Author response to Decision Letter 1]

23 Dec 2024

[Aifeng Liu]

[First Teaching Hospital of Tianjin University of Traditional Chinese Medicine, Tianjin 300381, China]

[draifeng@163.com]

Editor-in-Chief,

PLOS ONE

Subject: Response to Reviewer Comments - Manuscript ID [PONE-D-24-53513]

Dear Editor-in-Chief,

We sincerely appreciate the valuable feedback provided by both the reviewers and the Editor for our manuscript titled " Association between age at first birth and risk of rheumatoid arthritis in women:evidence from NHANES 2011–2020" (Manuscript ID: PONE-D-24-53513). We have thoroughly reviewed and carefully considered all the comments received, and we have made comprehensive revisions to the manuscript, with text changes highlighted in red, to enhance its quality and clarity. We have meticulously reviewed the reference list to ensure its completeness and accuracy. Please find below our point-by-point response to the comments received:

**Reviewer 1:**

[Minor comment 1] • Please correct the abstract on the title page of the manuscript also.

Response: Thank you very much for your review. We have corrected the abstract on the title page of the manuscript.

**Reviewer 2:**

[Comment 1] •I found the revision sufficient.The authors have positively evaluated my suggested contributions in their articles. The changes made are satisfactory to me.

Response:Thank you for your feedback. 

**Reviewer 3:**

[Comment 1] •1. Inconsistency in Claims about Previous Research

The introduction claims that no studies have specifically examined the relationship between Age at First Birth (AFB) and rheumatoid arthritis (RA) risk. However, the discussion section references several studies that indirectly or directly address this topic. This discrepancy undermines the novelty of the study and should be clarified.

Response: Thank you for your in-depth feedback on our study. We have made the necessary revisions to the introduction section of the manuscript.（Please refer to line 48 in the manuscript.）

[Comment 2] •2. Stylistic Deviations and Non-Academic Tone

The manuscript contains several instances of stylistic deviations from an academic tone, such as:

a) The phrase, "Policymakers must focus more on women in these AFB age ranges in screening RA and making preventive measures," employs emotionally charged language ("must") and shifts the tone from scientific to political. A more neutral phrasing is recommended.

b) A subjective critique of Zhu et al.'s study, emphasizing insufficient sample size without detailed justification.

c) Statements regarding the increased risk due to late childbirth ("longer period to accumulate exposures to environmental and lifestyle factors") presented as factual without sufficient evidence.

d) These examples are not exhaustive; the authors are encouraged to thoroughly review the manuscript for similar stylistic deviations to enhance its academic rigor.

Response: Thank you for your valuable feedback on the article. 

a) We have made the necessary revisions to the conclusion section of the manuscript.

（Please refer to line 34 in the manuscript.）

b) We have provided an explanation for the study conducted by Zhu et al.

（Please refer to line 143 in the manuscript.）

c)We have supplemented the discussion section of the article with relevant references to support our conclusions.

References：

[30]Winkler-Dworak M, Pohl M, Beaujouan E. Scenarios of Delayed First Births and Associated Cohort Fertility Levels. Demography. 2024 Jun;61: 687–710. doi:10.1215/00703370-11315685.

[31]Ritonja JA, Madathil S, Nicolau B, L’Espérance K, Ho V, Abrahamowicz M, et al. Body fatness across the adult life course and ovarian cancer risk. Eur J Epidemiol. 2024 Oct;39: 1139–1149. doi:10.1007/s10654-024-01161-1.

d)Thank you for your review. We have conducted a thorough examination of the entire manuscript to ensure its academic rigor.

[Comment 3] •3. Lack of Control for Multiple Comparisons and Type I Errors

The manuscript reports multiple statistically significant findings (P < 0.05) across various subgroups and categories but does not address whether adjustments for multiple comparisons were applied. This raises concerns about inflated Type I error rates (false positives). The absence of correction methods like Bonferroni or Holm adjustments undermines the validity of some conclusions, particularly given the extensive number of statistical tests performed. The authors should clarify whether such adjustments were applied or acknowledge this as a limitation in the discussion.

Response: We appreciate the reviewer’s rigorous suggestions regarding the statistical analysis in this manuscript. This feedback is highly valuable for improving both the current study and our future research. It is true that Bonferroni correction or other multiple comparison adjustments were not applied when analyzing variables with multiple subgroups, which represents a limitation of this study. However, most of the p-values for the variables included in our analyses were less than 0.001, indicating statistically significant differences with a very low probability of Type I error. Therefore, even without multiple comparison adjustments, the potential bias introduced in this study is minimal, and the overall impact on the conclusions is likely negligible.

Additionally, we note that other studies utilizing the NHANES database have similarly not universally applied Bonferroni correction for subgroup comparisons. Nevertheless, we recognize the importance of addressing multiple comparisons and will prioritize this aspect in future studies to further enhance the rigor and reliability of our research.

References：

[1]Paulose-Ram R, Graber JE, Woodwell D, Ahluwalia N. The National Health and Nutrition Examination Survey (NHANES), 2021-2022: Adapting Data Collection in a COVID-19 Environment. Am J Public Health. 2021 Dec;111(12):2149-2156. doi: 10.2105/AJPH.2021.306517. PMID: 34878854; PMCID: PMC8667826.

[2]Liu Z, Kuo PL, Horvath S, Crimmins E, Ferrucci L, Levine M. A new aging measure captures morbidity and mortality risk across diverse subpopulations from NHANES IV: A cohort study. PLoS Med. 2018 Dec 31;15(12):e1002718. doi: 10.1371/journal.pmed.1002760. PMID: 30596641; PMCID: PMC6312200.

We are confident that these revisions have substantially enhanced the manuscript and brought it in line with the standards set by PLOS ONE.

Enclosed with this response letter, please find the updated manuscript, including all necessary changes and tracked revisions. We believe that these modifications effectively address the feedback received.

At the same time, we would like to thank the reviewers and the editor for their constructive feedback and guidance, which have greatly contributed to the refinement of our work.

We appreciate the opportunity to revise and resubmit our manuscript, and we are hopeful for a positive outcome. We remain committed to addressing any further suggestions or recommendations from the reviewers and the Editor. Thank you for your consideration.

Sincerely,

Aifeng Liu

---

## [Decision Letter · Decision Letter 2]

30 Dec 2024

Association between age at first birth and risk of rheumatoid arthritis in women: evidence from NHANES 2011–2020

PONE-D-24-53513R2

Dear Dr. Liu,

We’re pleased to inform you that your manuscript has been judged scientifically suitable for publication and will be formally accepted for publication once it meets all outstanding technical requirements.

Kind regards,

Ahmed Qasim Mohammed Alhatemi

Academic Editor

PLOS ONE

Additional Editor Comments (optional):

Reviewers' comments:

Reviewer's Responses to Questions

**Comments to the Author**

1. If the authors have adequately addressed your comments raised in a previous round of review and you feel that this manuscript is now acceptable for publication, you may indicate that here to bypass the “Comments to the Author” section, enter your conflict of interest statement in the “Confidential to Editor” section, and submit your "Accept" recommendation.

Reviewer #1: All comments have been addressed

Reviewer #2: All comments have been addressed

Reviewer #3: All comments have been addressed

2. Is the manuscript technically sound, and do the data support the conclusions?

Reviewer #1: Yes

Reviewer #2: Yes

Reviewer #3: Yes

3. Has the statistical analysis been performed appropriately and rigorously? 

Reviewer #1: Yes

Reviewer #2: No

Reviewer #3: Yes

4. Have the authors made all data underlying the findings in their manuscript fully available?

Reviewer #1: Yes

Reviewer #2: Yes

Reviewer #3: Yes

5. Is the manuscript presented in an intelligible fashion and written in standard English?

Reviewer #1: Yes

Reviewer #2: Yes

Reviewer #3: Yes

6. Review Comments to the Author

Reviewer #1: All my concerns have been addressed by the authors. I have no more queries

All the best for your article.

Reviewer #2: From my perspective, the necessary changes that I suggested have been made. I have no further reccomendation.

Reviewer #3: After a thorough second review of the manuscript, I am pleased to confirm that the article meets the necessary standards for publication in a scientific journal. The authors have addressed the previous concerns effectively, improving the clarity, structure, and scientific rigor of the paper.

The study presents a well-conceived methodology, sound statistical analysis, and a clear discussion of the findings, which contribute significantly to the field. Furthermore, the adjustments made to ensure consistency in tone and academic style enhance its readability and impact.

I recommend the manuscript for publication, as it represents a valuable contribution to the scientific community.

7. PLOS authors have the option to publish the peer review history of their article (what does this mean?). If published, this will include your full peer review and any attached files.

Reviewer #1: No

Reviewer #2: No

Reviewer #3: **Yes: **Artyom Borbat

---

## [Editor Report · Acceptance letter]

8 Jan 2025

PONE-D-24-53513R2 

PLOS ONE

Dear Dr. Liu, 

I'm pleased to inform you that your manuscript has been deemed suitable for publication in PLOS ONE. Congratulations! Your manuscript is now being handed over to our production team.

Kind regards, 

on behalf of

Dr. Ahmed Qasim Mohammed Alhatemi 

Academic Editor

PLOS ONE